# Simulation Study for Hydraulic Fracture Monitoring Based on Electromagnetic Detection Technology

**Liming Zhang [1,\*], Xingyu Zhou [1], Jijia Sun [2], Peiyin Jiang [1], Zhihao Lu [1] and Cheng Cheng [1]**

[1] School of Petroleum Engineering, China University of Petroleum (East China), Qingdao 266580, China
[2] Oil and Gas Technology Research Institute of Changqing Oilfield Branch of CNPC, Xi'an 710018, China
\* Correspondence: zhangliming@upc.edu.cn

**Abstract:** The stimulated reservoir volume (SRV) technology extends conventional fracturing technology. Understanding how to effectively and accurately determine modified fracture shape and volume is the key point to evaluating the stimulation effect. Using electromagnetic detection technology can provide a new option for measuring these parameters. By the finite method, the rationality of electromagnetic detection technology to obtain the relevant parameters of reconstruction fracture is testified through forward simulation. This study compared the signals of fractures with different conductivity, volume, and shape collected by electromagnetic detection tool, and the results show that the signals have a specific correspondence with fracture geometric parameters. According to the electromagnetic signal curve of the forward model, the description of propped fractures, including positions and sizes, can be realized.

**Keywords:** hydraulic fracturing; electromagnetic detection; finite element method





## 1. Introduction

Hydraulic fracturing technology is widely used in oil gas exploitation, which can generate new fractures to obtain access to unconventional oil and gas [1–3]. To evaluate the effect and performance of hydraulic fracturing, it is necessary to accurately monitor and quantify the fractured support at any time [4–6]. Micro-seismic monitoring and inclinometer logging are two kinds of propped fracture monitoring techniques widely used in oil fields [7–10]. However, both are based on the physical events during proppant propagation to achieve fracture monitoring and cannot measure events correlated with proppant settlement distribution [11–14]. Therefore, neither method can monitor proppant distribution and effective fracture information in the hydraulic fracture [15,16]. In addition, the common potential logging method is susceptible to the influence of edge and bottom water in the reservoir, and this technology has a high cost [17,18]. A more promising alternative is to use electromagnetic detection technology to effectively monitor and evaluate hydraulic fracturing propped fractures [19,20], its basic principle is to obtain the induction signal of the fractures at different positions, and the simple schematic is shown in Figure 1.

The author of [21] proposed in 2012 that ferrofluid injection into fracture sup-42 port fracture makes fracture become a magnetic medium with electromagnetic properties 43 significantly different from the background reservoir. According to Faraday's law of electromagnetic induction, fractures that become magnetic media will be magnetized under the action of a time-varying magnetic field and generate eddy currents. The resulting eddy currents further generate a secondary magnetic field. Because there is a phase difference between the initial and secondary magnetic fields, the electromagnetic signals collected by inversion can be used to obtain the geometric parameters of the propped fractures.

In the induction tool domain, the author of [22] used the antenna resonance and low-frequency induction models to verify the feasibility of electromagnetic detection of fractures filled with conductive proppant. The authors of [23] designed a low-frequency induction

logging tool consisting of a three-way emitter and three receivers spaced differently from the emitter to enable electromagnetic monitoring of conductive proppant-packed frac fractures. This method has the potential to estimate the propped length, height, and orientation of hydraulic fractures and can also estimate the vertical distribution of proppant within the fracture. The authors of [24,25] designed and constructed an electromagnetic induction tool consisting of coaxial, coplanar, and cross-polarized triaxial transmitters and receiving coils. The induction tool with a coaxial coil structure has a strong sensitivity to crack surface area and dip angle. The induction tool with a coplanar structure is sensitive to fracture length–diameter ratio.

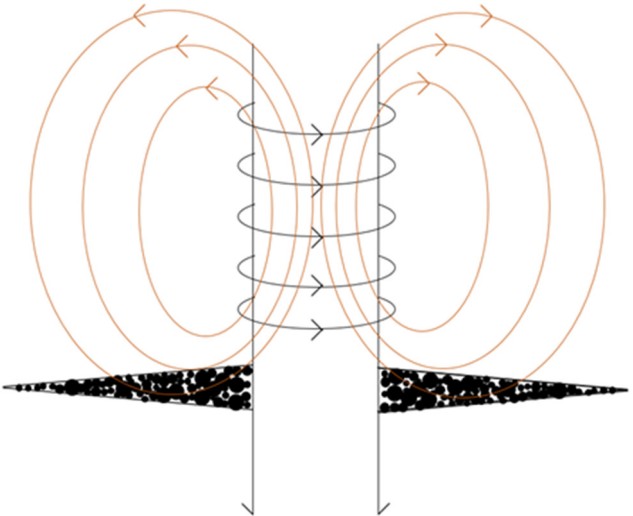

**Figure 1.** Simple schematic diagram of electromagnetic monitoring.

In the numerical modeling domain, the authors of [26] used numerical simulations to discuss and evaluate the feasibility of using magnetic field measurements to monitor propped fractures. It is found that characteristic parameters such as propped fracture depth, total volume, and magnetic susceptibility difference of injected magnetic proppant can be obtained by processing abnormal electromagnetic signals with an appropriate inversion algorithm. The authors of [27] used the commercial finite element software ANSYS Maxwell to study the effect of the conductivity, opening, length, inclination angle, and distance from the wellbore to the propped fracture in a vertical well on the response of the resistivity induction log signal. The authors of [28,29] conducted a numerical simulation study on electromagnetic fracture monitoring by combining the stable double conjugate gradient fast Fourier transform method with the numerical mode matching method (NMM-BGS-FFT).

In this paper, a simplified forward model is established by using COMSOL Multiphysics software (Version 5.6). In the design of the model, the transmitter and receiver of the HDIL detection tool (Western Atlas International, Huntington, CA, USA) are simplified to magnetic dipoles, and hydraulic fractures are reduced to geometry with only electromagnetic properties. The whole simulation obtains the response of the magnetic dipole to the fractures at different positions. Then, the signal reception performance of the probe and the influence of the formation on the signal were evaluated by analyzing the voltage signal under the different reservoir and magnetic dipole parameters. Finally, by changing the cross-section geometry size of the fracture, we obtained the law of fracture shape and geometry size on electromagnetic signal response.

## 2. Mathematics Physics Model

According to the process of electromagnetic monitoring of hydraulic fractures, the work involves four main parts, horizontal wells, background formations, electromagnetic detection tools, and effective fracture propping with conductive proppant. Horizontal wells are simplified as open-hole wells with vertical and horizontal sections interconnected.

The background formation is simplified as a cuboid with homogeneous anisotropy. More specifically, the calculation area will be reduced to the area near the propped fracture, and the cuboid area of the background formation will cover all elements, including propped fractures, detection tools, and the horizontal wellbore. At the same time, we set an infinite meta-field outside the cuboid background formation to ensure the calculation accuracy when reducing the calculation area of the model.

We choose high-definition induction logging (HDIL) as the model of detection tools which is a common tool for induction logging [30]. Its structure is shown in Figure 2. The HDIL probe tool includes eight operating frequencies (10–150 kHz) and seven three-coil line subarrays [31]. Table 1 shows the related parameters of each subarray of the HDIL detection tool. In the actual modeling process, the transmitting coil of the HDIL detection tool is simplified as a magnetic dipole, which is represented in geometry as a point in space. The shielding and receiver coils are simplified as hollow cylinders with opposite winding directions [32,33].

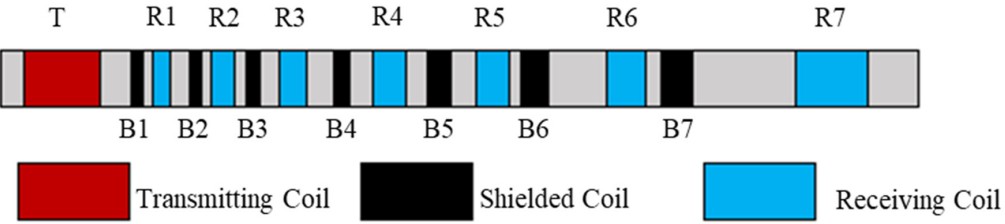

**Figure 2.** Structure diagram of HDIL detection tool.

**Table 1.** Subarray parameters for HDIL detection tool.

| Subarray | | 1 | 2 | 3 | 4 | 5 | 6 | 7 |
|---|---|---|---|---|---|---|---|---|
| Pickup dipole | Turns | 12 | 21 | 33 | 51 | 82 | 128 | 200 |
| | Spacing/m | 0.152 | 0.254 | 0.399 | 0.622 | 0.978 | 1.524 | 2.388 |
| Shielded coil | Turns | −6 | −10 | −15 | −21 | −31 | −47 | −70 |
| | Spacing m | 0.121 | 0.198 | 0.307 | 0.463 | 0.707 | 1.091 | 1.683 |

Regarding material setup, the formation is regarded as a homogeneous isotropic medium and sets the formation to a uniform electromagnetic property. The propped fracture area is assumed to be filled with magnetic proppant, so this area is set to the electromagnetic properties of the conductance proppant [34]. Moreover, the material properties of the wellbore area are set to electromagnetic properties similar to those of conventional fracturing fluids.

After defining the geometry, parameters, materials, mesh building, and other conditions of all elements, taking the coil as the research object, perform the parametric sweep according to the path of a wellbore [35]. The step is set to 0.1 m. Parametric sweep mainly includes coil geometry analysis and frequency domain analysis.

### 2.1. The Boundary Conditions

In realizing electromagnetic monitoring of fracture support, the electromagnetic monitoring tool can generate electromagnetic waves by its alternating current, and the research area mainly includes an electrostatic, electric current, and magnetic field. On the other hand, the whole system has an identical frequency, and the electromagnetic responses vary sinusoidally [36]. Based on it, the finite-element simulation is implemented by the frequency domain simulation in COMSOL. The governing equations are as follows:

$$B = \nabla \times A \tag{1}$$

$$\nabla \times \left( \mu^{-1} \nabla \times A \right) + \left( j\omega\sigma - \omega^2\varepsilon \right) A = 0 \tag{2}$$

where $A$ is the magnetic vector potential, Wb/m; $B$ is the magnetic induction, T; $\mu$ is the permeability, H/m; j is the square root of $-1$; $\omega$ is the angular frequency, rad/s; $\sigma$ is the electrical conductivity, S/m; $\varepsilon$ is the dielectric constant.

After determining the governing equations, the boundary conditions and radio source should be determined. Firstly, the transmitting coil of the HDIL probe tool should be simplified into a magnetic dipole [37,38], and its magnetic dipole moment is $M_S$:

$$M_S = I_T N_T p r_T^2 \tag{3}$$

where $M_S$, J/T; $I_T$ is the current of the transmitting coil, $A$; $N_T$ is the number of turns; $r_T$ is the radius of the transmitting coil, m.

A perfect magnetic conductor means the tangential magnetic flux density equal to zero, and magnetic insulation means the normal magnetic flux density equal to zero. Here the equations of them are given as follows:

$$n \times H = 0 \tag{4}$$

$$n \times A = 0 \tag{5}$$

Beyond the computational domain, we set an additional infinite domain. Hence, no other boundary conditions were required.

For the wellbore orientation, the strength of the magnetic field at a point away from the transmitting coil is $H_z$:

$$H_Z = \frac{M_S}{2\pi z^3}(1 + ik_i z)e^{-ik_i z} \tag{6}$$

$$k_i = \omega\sqrt{\frac{M_S}{2\pi z^3}} \cdot \sqrt{1 + \sqrt{1 + \left(\frac{\sigma}{\omega\mu}\right)^2}} + i\omega\sqrt{\frac{M_S}{2\pi z^3}} \cdot \sqrt{\sqrt{1 + \left(\frac{\sigma}{\omega\mu}\right)^2} - 1} \tag{7}$$

where $z$ is the distance between a point in the wellbore direction and the emission source. $k_i$ can be reduced to $k_i = \beta + i\alpha$. $\beta$ is phase constant, $\alpha$ is the attenuation constant.

### 2.2. Mesh Generation

Volume meshes are usually generated using structured or unstructured grids composed of either tetrahedral or hexahedral elements [39]. For this model, we need to produce smaller grids in fracture and wellbore areas, and rougher meshes are produced in other areas not covered by our study. For the infinite domain of formation, which was conducted as a 3D model, sweeping was used to build the hexahedral mesh, as shown in Figure 3. For the shaft and hydraulic fractures domain, mapping was used to build the tetrahedral mesh, as shown in Figure 4. After this, the infill density of the transition zone of different domains should be set higher to improve calculation accuracy [40–42].

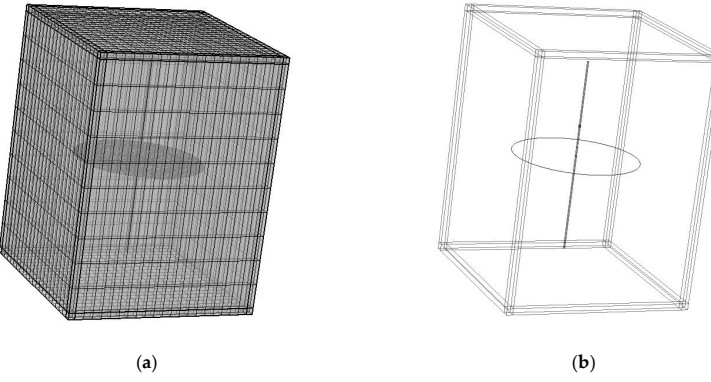

(a)　　　　　　　　　　　(b)

**Figure 3.** Computing domain diagram: (**a**) overall meshing diagram (hexahedral mesh); (**b**) model line drawing.

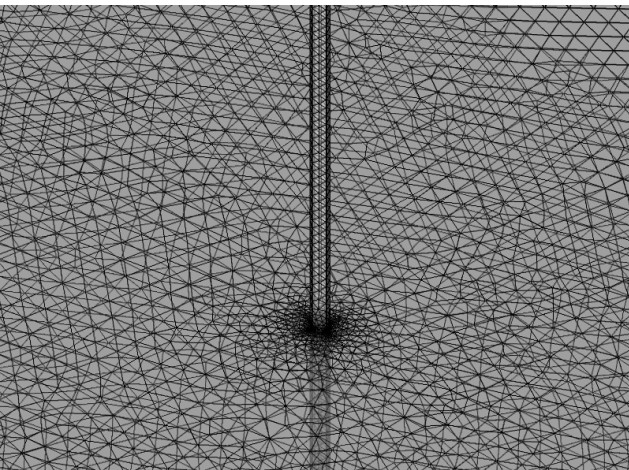

**Figure 4.** Meshing diagram of the shaft and hydraulic fracture (tetrahedral mesh).

## 3. Validation Experiments

### 3.1. Electromagnetic Signal Response Analysis

The purpose of establishing the forward mode of electromagnetic monitoring is to obtain the electromagnetic signal in different support layer cases, and according to the obtained different signal data, obtain the law between fracture parameters and electromagnetic signals. Firstly, we set an example to contrast the data on whether the fractures were filled with magnetic proppant. This example confirmed the feasibility of the proposed design and verified the sensitivity of the probe to changes in formation conditions. Then, we need to vary the basic model parameters, including the conductivity of the background formation, the conductivity of the proppant, and emission frequency.

During the next three sensitivity analyses experiment, the relative dielectric constant and the relative magnetic permeability were set to 1. We set the fracture position to 0 m and define the fracture perpendicular to the wellbore as an orthometric fracture. The HDIL detection tool was moved from a position of 5 m to a position of −5 m in the horizontal wellbore. Data were acquired every 0.1 m as the tool moved. The working frequency of the detection tool is 10 kHz. In this work, we did not consider the effects of the well casing, so we set the electrical conductivity of the wellbore area to 1 S/m. A cylinder with a diameter of 5 m and a height of 1 mm was set as the monitoring subject.

#### 3.1.1. Electromagnetic Signal Response with and without Proppants

In this work, the signals most concerned with were the voltages detected by the receiving coils in the subarrays, including the real and imaginary parts.

When there is no propped fracture in the model, the real part of the voltage signal of each subarray are smooth lines with faint fluctuations, as shown in Figure 5a. In addition, as the ordinal number of the subarray increases (mean the distance between the transmitting source and the receiving coil increase), the amplitude of the voltage real part signal will decrease with it, and the distance between the amplitude of the voltage real part signal of the adjacent subarrays will gradually decrease. When it comes to subarray 4, the distance is very small, and all curves tend to be consistent. When there are fractures in the model, each subarray's voltage real part curves are symmetrical, and there will be an increase in the range signal near the fracture area. When there is propped fracture in the model, a signal mutation can be discovered in the fracture position, as Figure 5c shows. This signal mutation is generally present in all cases with propped magnetic fractures, and the real part signal increased as the serial number of subarrays increased. As for the imaginary part, its change rule is consistent with the real part. However, as Figure 5b,d shows, its amplitude is three orders of magnitude bigger than the amplitude of the real part, but for different subarrays, the amplitudes of the real and imaginary parts are fixed, as Figure 5a,b shows.

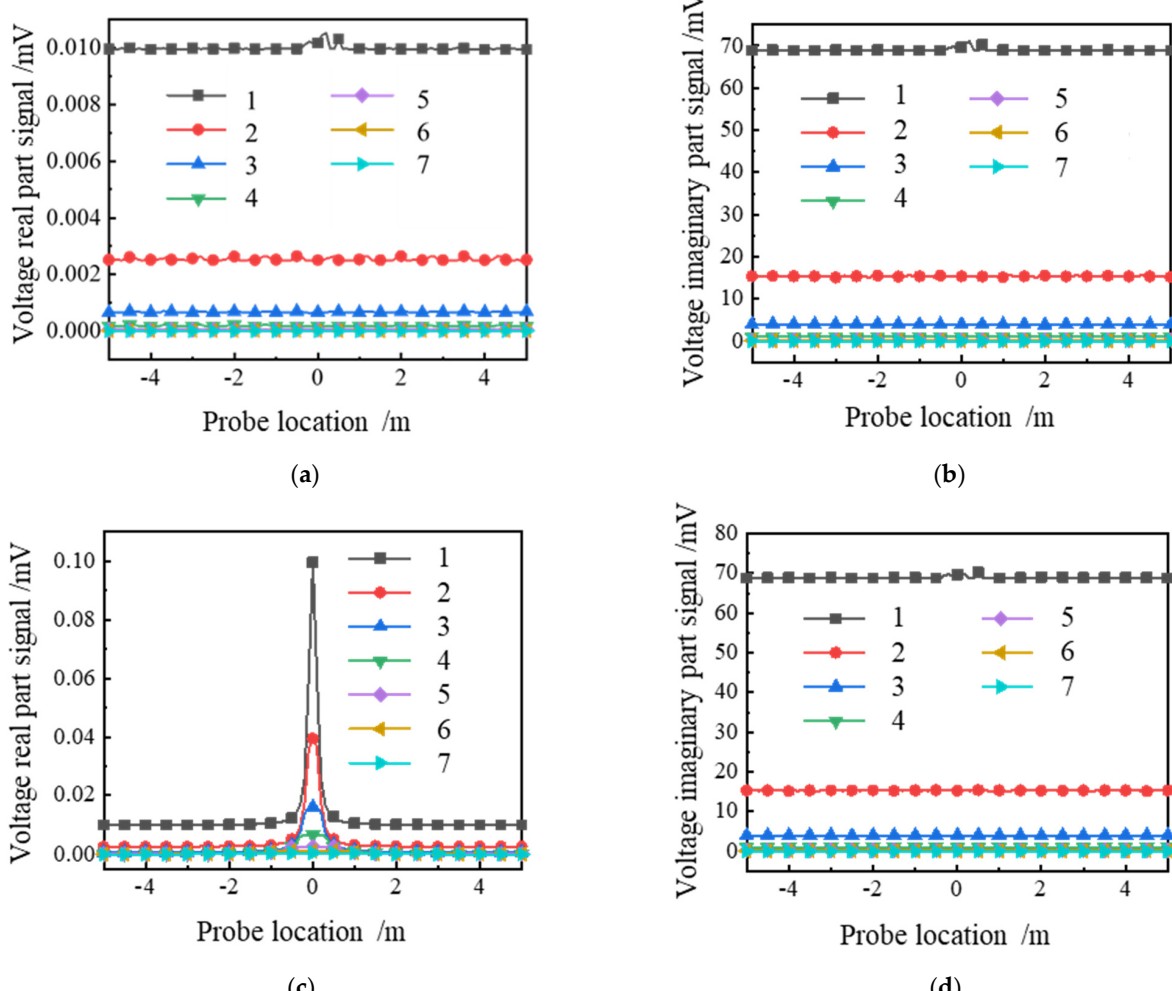

**Figure 5.** Electromagnetic signal response curves with and without proppant: (**a**) real part signal without fracture; (**b**) imaginary part signal without fracture; (**c**) real part signal without fracture; (**d**) imaginary part signal without fracture.

### 3.1.2. Formation Conductivity

As confirmed above, the HDIL (Western Atlas International, Houston, TX, USA) probe has shown the ability to monitor fractures in magnetic proppant packs. However, the premise of the above simulation is that the formation conductivity remains constant, but in fact, the formation generally contains many magnetic materials, which causes errors to creep in when the formation conductivity increases. Therefore, we need to change the background electrical conductivity to observe its influence on the signal amplitude.

Under the same fracture electrical conductivity condition, respectively, set the formation conductivity to 0.001 S/m, 0.01 S/m, 0.1 S/m, and 1 S/m. Then, the voltage signal of subarray 1, subarray 4, and subarray 7 would be obtained every 0.1 m from 5 m to −5 m. The voltage signal charts are shown in Figure 6. The real part of the voltage increases with the formation conductivity, as Figure 6a,c,e show. However, the increase in signal amplitude caused by magnetic proppant does not vary with formation conductivity. Thus, it can be conducted that formation conductivity only affects the voltage signal in the non-fractured propped fracture area but has little effect on the voltage signal in the propped fracture area. Analyzing Figures 5c and 6e, we found that the signal from subarray 7 is not missing, but the overall amplitude decreases and the shape of the abnormal signal also changes. By comparing Figure 6a,c,e, we found that the shape of the amplitude would change with the subarray. The longer separation is, the wider the response area. This indicates that the voltage amplitude is just related to the number of turns in the receiving

coil and the conductivity of the mediums. The voltage imaginary part signals of the three groups of subarrays do not show obvious overall change law with the increase in formation conductivity, and the curve is unstable fluctuation. When the separation of subarrays increased, the voltage imaginary part signal curves under different formation conductivity would gradually divide, and the larger the conductivity of the formation, the lower the amplitude of the imaginary part of the voltage.

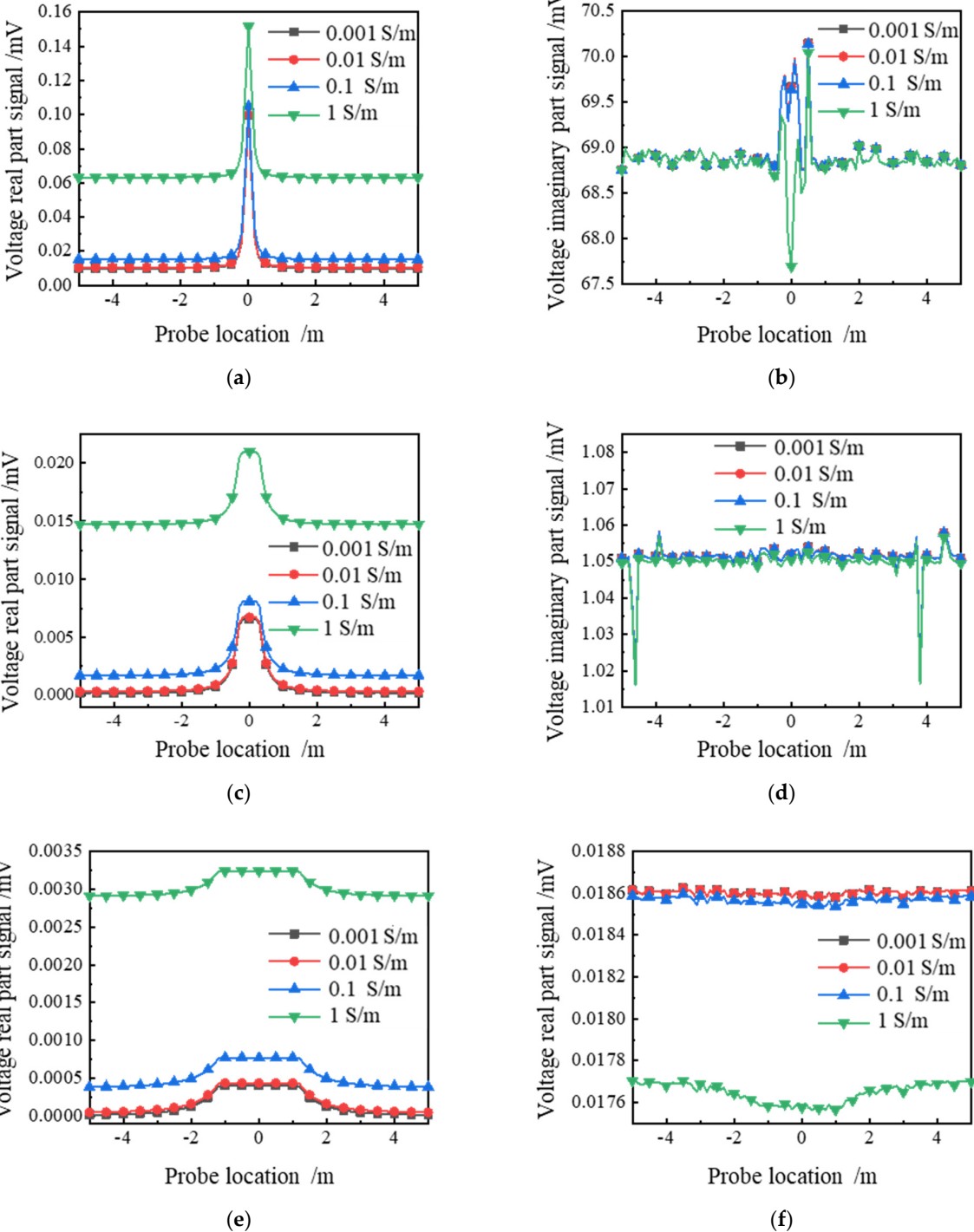

**Figure 6.** Electromagnetic signal response curves under different formation conductivities: (**a**) real part signal of subarray 1; (**b**) imaginary part signal of subarray 1; (**c**) real part signal of subarray 4; (**d**) imaginary part signal of subarray 4; (**e**) real part signal of subarray 7; (**f**) imaginary part signal of subarray 7.

### 3.1.3. Effect of Probe Frequency

Under the condition that the conductivity of the fracture and the background formation remain unchanged, by changing the working frequency of the electromagnetic detection tool, several groups of electromagnetic signals at different frequencies would be obtained. Analyzing these electromagnetic signals can explore the influence of probe frequency on the monitoring effect.

Figure 7 shows the real part voltage signal and imaginary part voltage signal of subarray 1, subarray 4, and subarray 7 under eight operating frequencies of 10–150 kHz. With the increase in operating frequency electromagnetic detection tools, areas of the three subarrays of real voltage signal strength gradually enhanced. Moreover, every additional 20 kHz frequency caused the voltage real part signal increases further in the vicinity of the propped fracture zone. Accordingly, the voltage real part signal near the propped fracture area is highly sensitive to the operating frequency of the electromagnetic detection tool, as Figure 7a,c,e show. The amplitude of the voltage imaginary part signal curve of the three groups of subarrays is also significantly different with the change in operating frequency, as Figure 7b,d,f show. When the working frequency is larger, the amplitude of the electromagnetic imaginary part signal curve is larger. However, the overall trend of the voltage imaginary part signal curve did not change significantly. In contrast, the voltage imaginary part signal near the propping fracture area of the fracture was slightly decreased in the high operating frequency of subarray 7. Therefore, each subarray's voltage imaginary part signal also has a strong sensitivity to the change in operating frequency.

### 3.1.4. Fracture Conductivity

Under the condition that the conductivity of the fractured fracture and the conductivity of the background formation remains unchanged, we wound to change the electrical conductivity of the fracture, which is filled with conductive proppant. Moreover, compare their voltage real signal and voltage virtual signal.

Figure 8 shows the real and imaginary voltage signals of the receiving coils of 10 S/m, 100 S/m, 1000 S/m, and 1000 S/m fractures under subarray 1, subarray 4, and subarray 7. With the increased conductivity of the propped fracture, the overall amplitude of the mutation signal curve of the three subarrays will increase continuously. Moreover, the amplitude difference of the real part of the voltage is very large in every order of magnitude. This indicates that different types of conductance proppants have a strong influence on the voltage real part signal of each subarray, and the strength of the voltage real part signal will increase with the increase in the voltage real part signal, indicating that voltage real part signal is highly sensitive to the conductivity of conductance proppant. On the contrary, in fracturing propping fractures with different electrical conductivity, the voltage imaginary part signal curve of subarray 1 presents a chaotic distribution and does not show a clear change rule. However, the imaginary part signal of subarray 4 was able to distinguish the crack with a conductivity of 10,000 S/m. The imaginary part signal of subarray 7 identified fractures with conductivity above 1000 S/m. By comparing the imaginary part signals in Figures 6 and 8, it is found that the imaginary part signal only produces a strong response to the material with high conductivity.

### 3.2. Single Fracture Forward Model

The research content of this chapter used the aforementioned finite element analysis method, which established a few forward models of electromagnetic monitoring for numerical simulation and then used single factor analysis to find the correlation law. Among them, the fracture's shape, size, and inclination angle are the main parameters studied in the forward modeling test. The influence of different geometric shapes is validated first using the circular, square, and square models. Then, validate whether the law of cross-section geometric size variation obtained from the orthogonal fracture model is also applicable to the inclined fracture model.

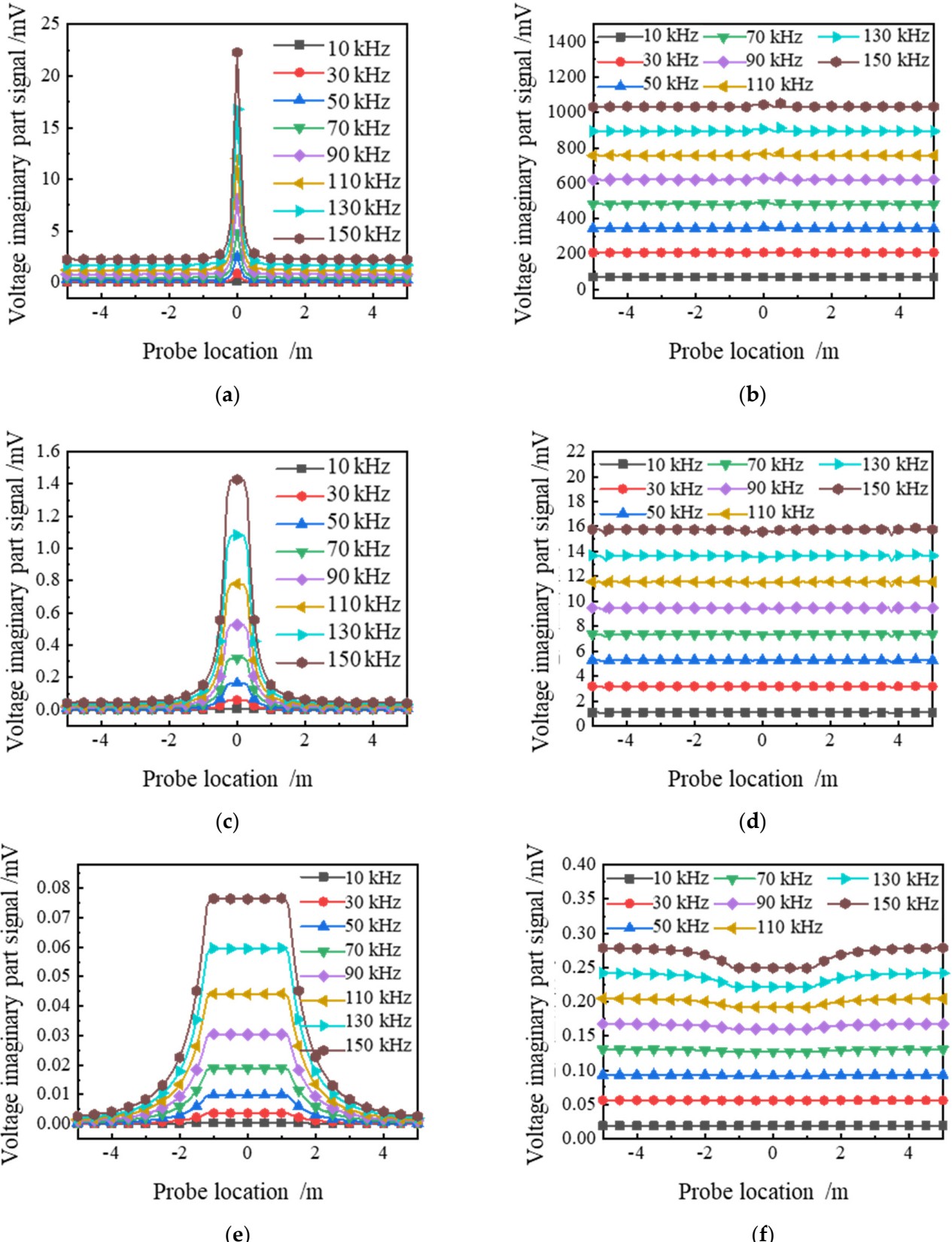

**Figure 7.** Electromagnetic signal response curves under different frequencies: (**a**) real part signal of subarray 1; (**b**) imaginary part signal of subarray 1; (**c**) real part signal of subarray 4; (**d**) imaginary part signal of subarray 4; (**e**) real part signal of subarray 7; (**f**) imaginary part signal of subarray 7.

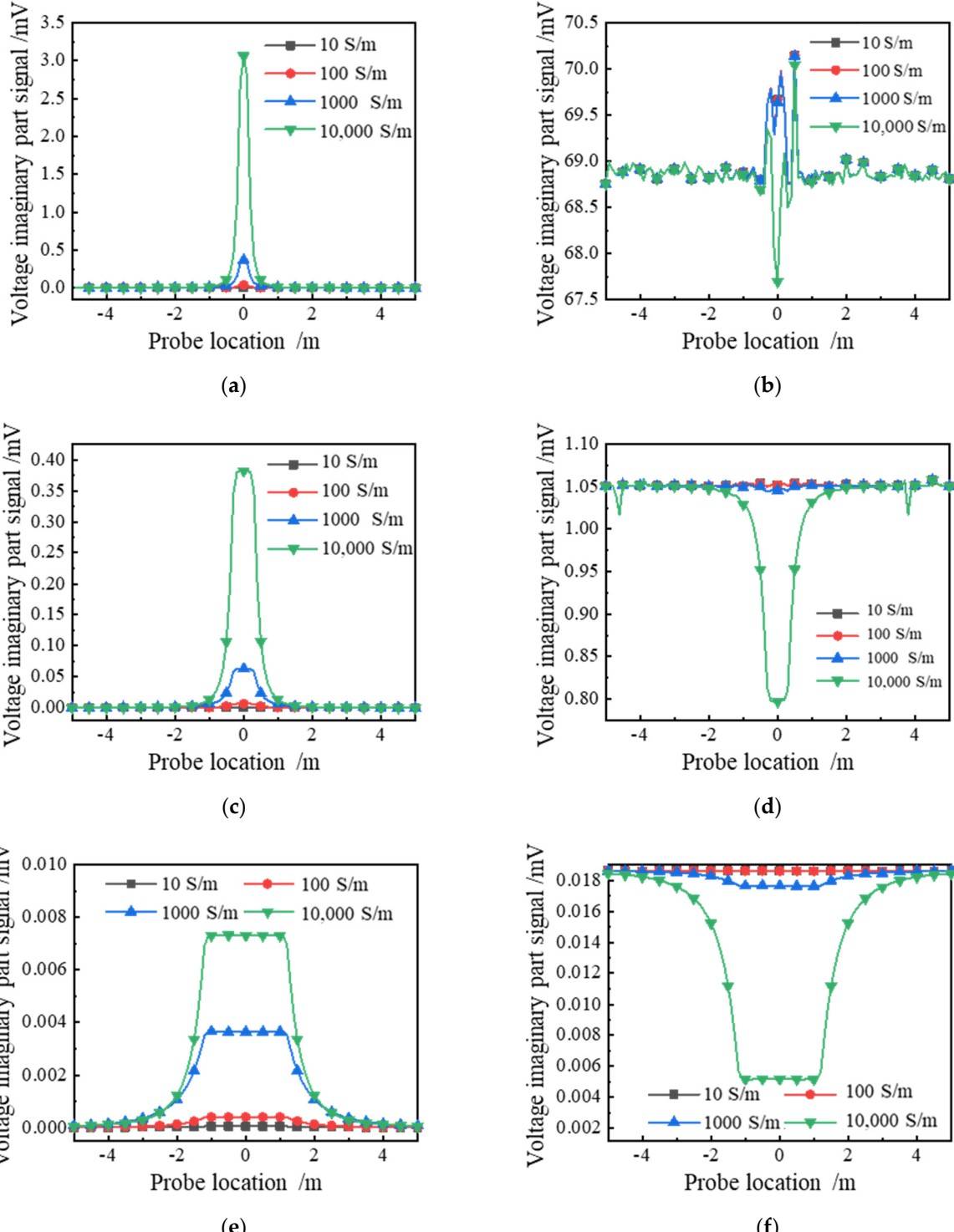

**Figure 8.** Electromagnetic signal response curves under different fracture conductivities: (**a**) real part signal of subarray 1; (**b**) imaginary part signal of subarray 1; (**c**) real part signal of subarray 4; (**d**) imaginary part signal of subarray 4; (**e**) real part signal of subarray 7; (**f**) imaginary part signal of subarray 7.

In order to simplify the calculation and present the results, the electromagnetic signal responses of representative subarrays 1, 4, and 7 are selected as the main results. Table 2 shows the unified model parameters of forward models at all levels.

**Table 2.** Forward modeling parameter table.

| Parameter Name | Formation Conductivity | Fracture Conductivity | Fracture Thickness | Wellbore Conductivity | Wellbore Radius | Operating Frequency of the Probe | Transmitting Coil Turns | Working Current | Coil Radius |
|---|---|---|---|---|---|---|---|---|---|
| Parameter Values | 0.001 S/m | 100 S/m | 0.005 m | 1 S/m | 5 in | 10 kHz | 50 | 1 A | 0.03 m |

### 3.2.1. Circular Model

We set the diameter of circular fracture models as 1 m, 10 m, and 100 m, which stand for micro, medium, and large models, respectively. Figure 9 shows a circular model with a diameter of 10 m.

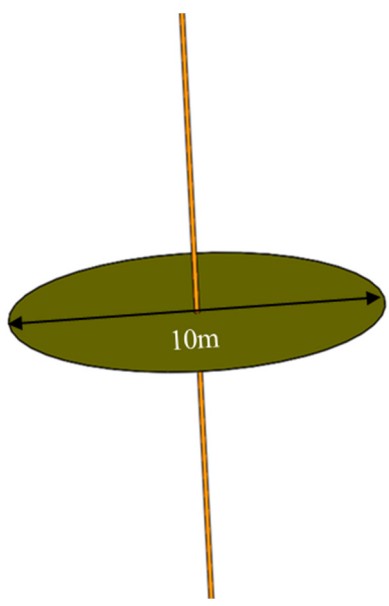

**Figure 9.** A circular fracture model with a diameter of 10m (The orange line is the wellbore).

Figure 10 shows the electromagnetic signal response curves of subarray 1, subarray 4, and subarray 7 of the circular model with three sizes. It can be found that, with the increase in subarray number, the signal of fracture location increases, and others decrease. In addition, the curves show different shapes at the fracture location. The mechanism of this can be explained from two aspects. Firstly, the greater the number of subarrays, the greater distance between the transmitting source and the receiving coil, which causes the smaller signal of the main electromagnetic wave received by the receiver coil. This contributes to the signal in the area except for fracture location decrease when the subarray number increases. Secondly, according to the preceding, with the increase in subarray number, the subarray real signal curve of the voltage amplitude will increase, and the area of anomalous variations will be widened. On the other hand, for the imaginary part signal shown in Figure 10b,d,f, the curve shape, amplitude, and trend did not change significantly with the increase in circular crack diameter.

To further study the influence of the geometric size of the propped fracture on the voltage signal of the circular fracture model, we simulated and calculated the real voltage signal of a circular crack model with a diameter of 0.5–100 m, as shown in Figure 11. For the circular fracture model, with the increase in diameter, the amplitudes of the three subarrays mutation signal curves all increased to different degrees. As Figure 11a shows, the overall voltage signal curve shape of subarray 1 does not change much with the increase in circular crack diameter. However, unlike subarray 1, with the increase in the diameter of the circular cross-section, the amplitude, shape, and range of the voltage real part signal curves of subarray 4 and subarray 7 change greatly. When the diameter of the circular model is less than 3 m, the mutation signal curve of subarray 4 shows a concave shape with lower middle and higher sides. As the diameter of the circular cross-section increases, the trough

of the curve at 0 m of the origin increases accordingly. However, when the diameter of the circular fracturing model reaches or exceeds 10 m, the voltage signal curve of subarray 4 does not change significantly. That means the limit size of the circular model that can be recognized by subarray 4 is reached.

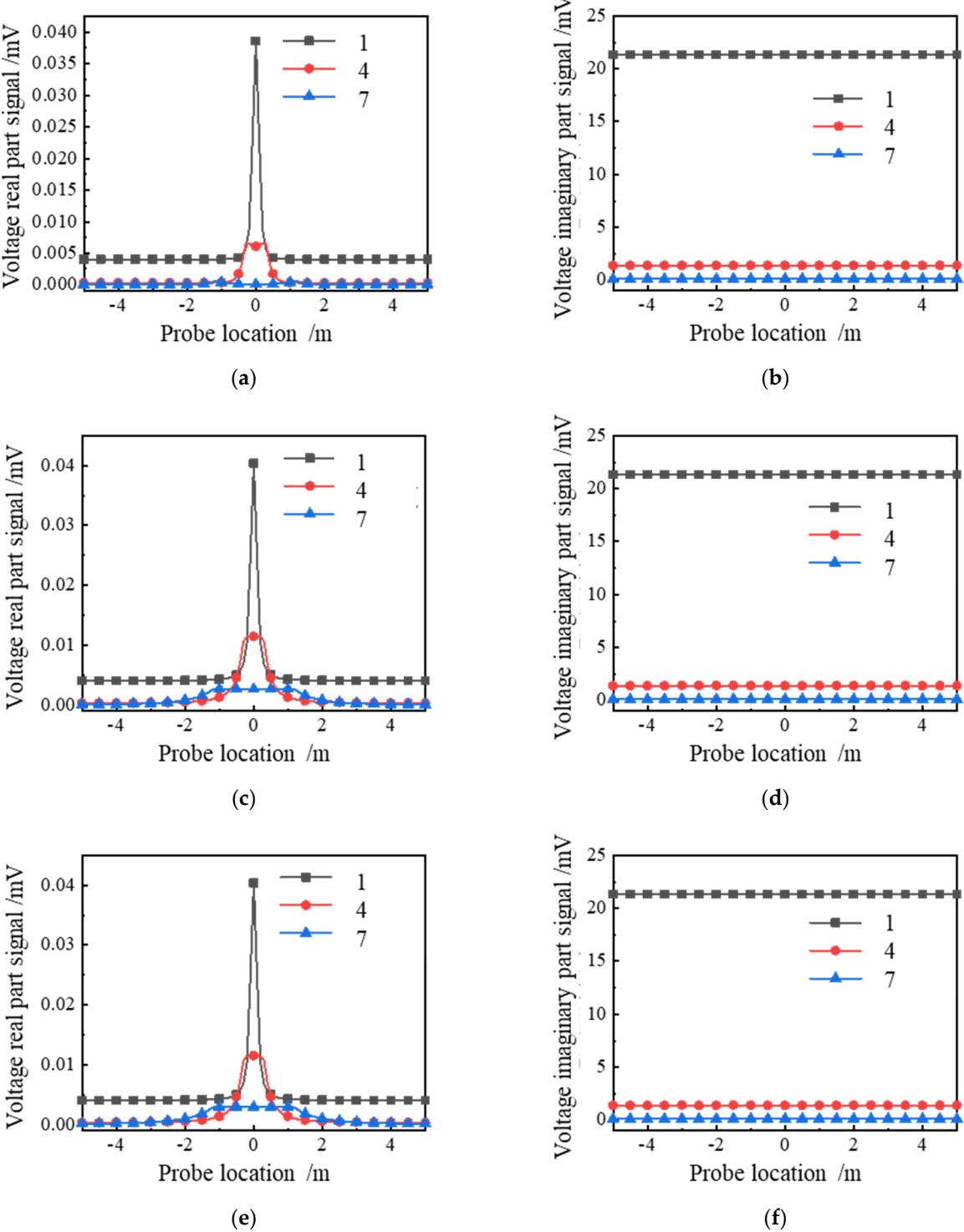

**Figure 10.** Electromagnetic signal response curves of circular fractures with different sizes: (**a**) real part signal of a 1 m diameter circular fracture; (**b**) imaginary part signal of a 1 m diameter circular fracture; (**c**) real part signal of a 10 m diameter circular fracture; (**d**) imaginary part signal of a 10 m diameter circular fracture; (**e**) real part signal of a 100 m diameter circular fracture; (**f**) imaginary part signal of a 100 m diameter circular fracture.

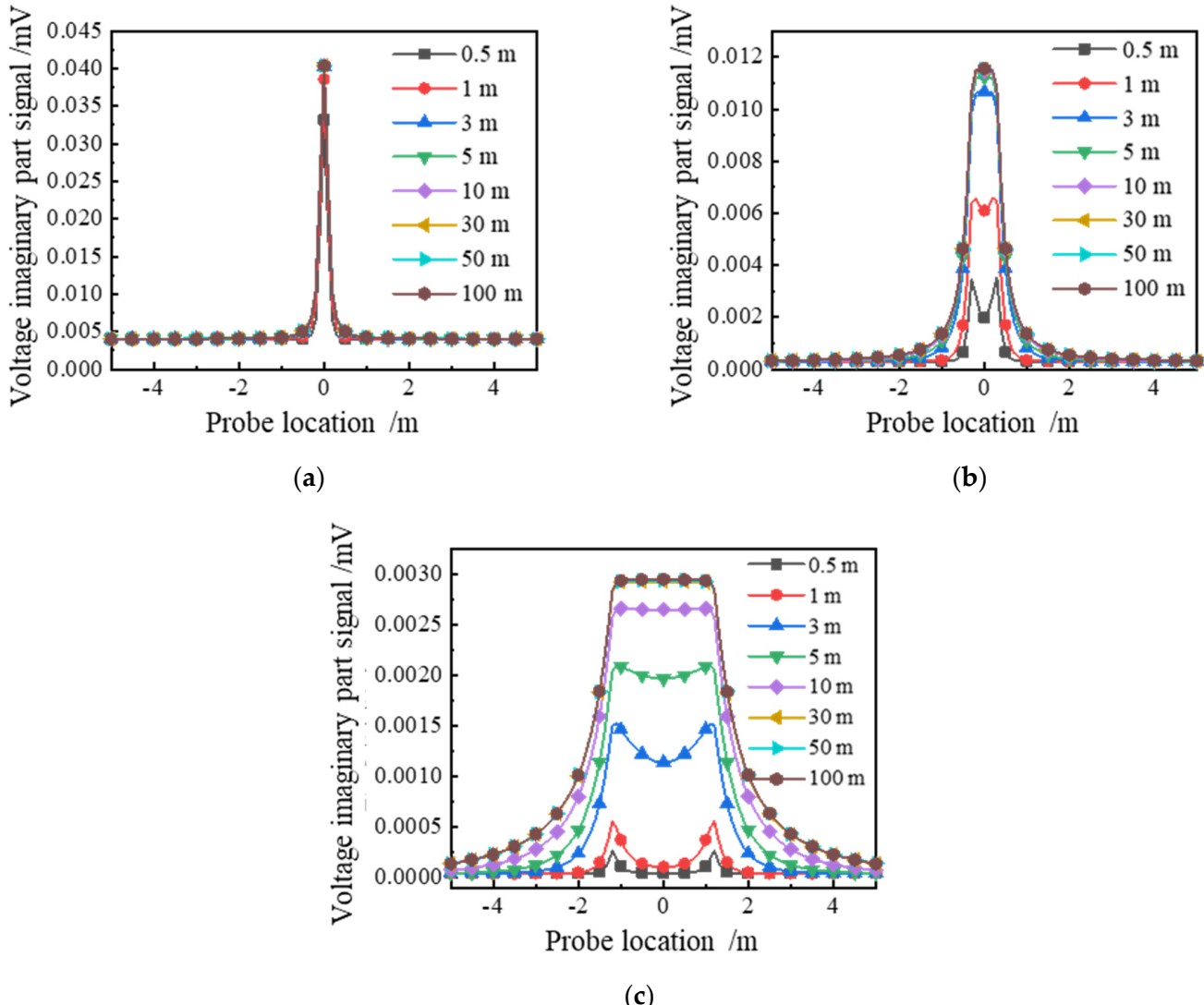

**Figure 11.** The voltage real part signal curves of circular fractures with different sizes: (**a**) real part signal of subarray 1; (**b**) real part signal of subarray 4; (**c**) real part signal of subarray 7.

The voltage real part signal curve of subarray 7 is similar to that of subarray 4, but it is better at identifying models larger than 3 m. Subarray 4 is better at about 1 m. This indicates that the coil system with different transmitting-receiving distances has different resolutions to fracture size. To further study its resolution, we performed a parameterized scan of fracture size for three subarrays from 0 m to 50 m (Figure 12). It can be found that the monitoring range of subarray 1 is within 5 m. The monitoring range of subarray 4 is within 10 m, and the monitoring range of subarray 7 is within 30 m.

### 3.2.2. Square Model

To study the influence of fracture shape on the monitored signal, we did the same work on the square model as the circular model, as Figure 13 shows. It is found that when the size of the square model is the same as that of the circular model, the real voltage signal curves of the square model and circular model are the same, and there is only a slight difference in amplitude. In contrast, the slight difference in amplitude is determined by the amount of conductive proppant.

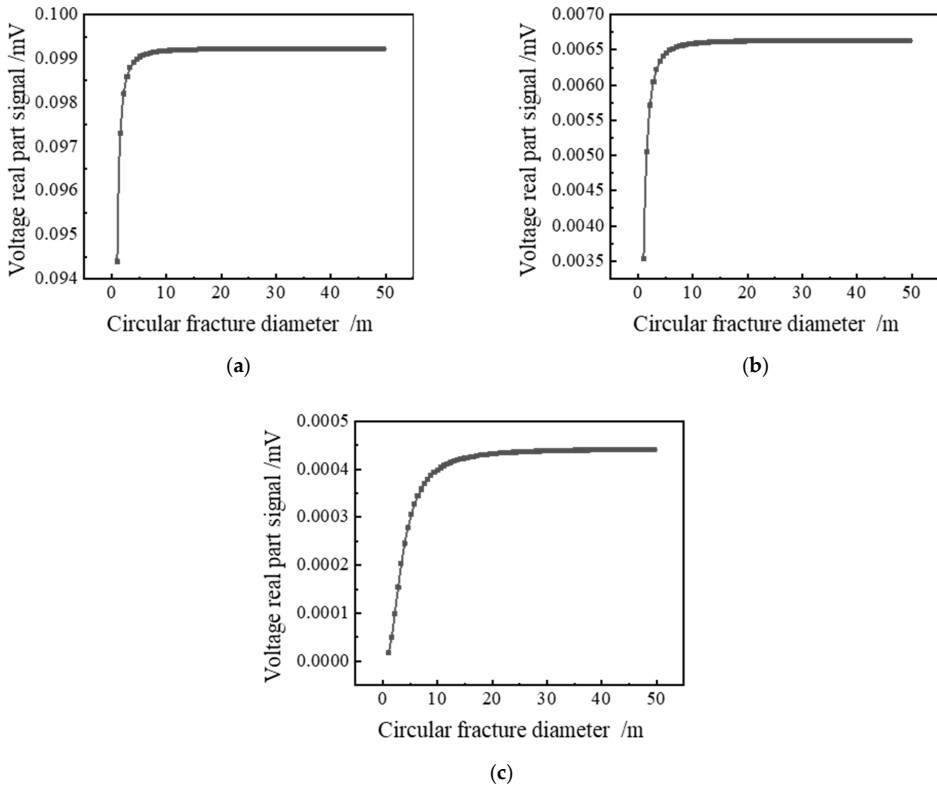

**Figure 12.** The relationship diagram between the amplitude of the real part of the voltage signal of the three groups' sub-arrays in origin and the diameter of the circular fracture: (**a**) real part signal of subarray 1; (**b**) real part signal of subarray 4; (**c**) real part signal of subarray 7.

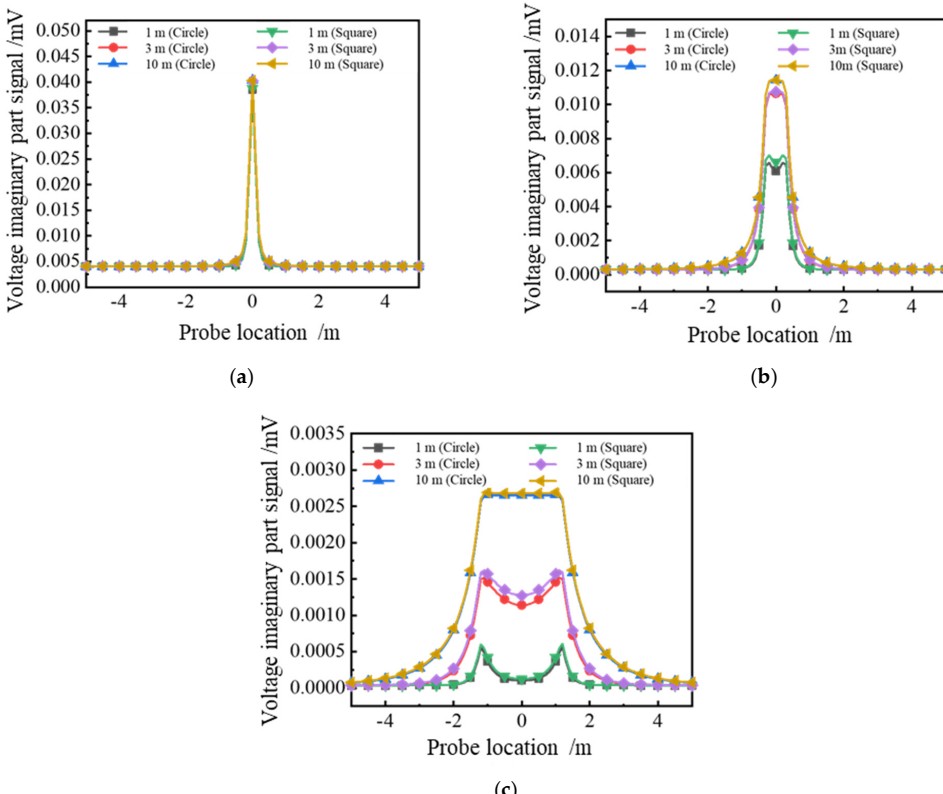

**Figure 13.** Voltage real part signal diagram between square fractures and circular fractures: (**a**) real part signal of subarray 1; (**b**) real part signal of subarray 4; (**c**) real part signal of subarray 7.

## 4. Discussion

By comparing the real part signal with the imaginary part signal, the real part signals reflect the presence or absence of proppant, which means the real part signal can better reflect the fracture characteristics. By comparing Figure 5a,c, the Probe can accurately detect the presence of fractures and locate them, and the magnitude of the amplitude is related to the transmitting-receiving distance. The law is that the larger the distance, the smaller the signal amplitude. The HDIL detection tool can monitor the otherness between fracture and formation.

Through comparative analysis, it can be seen that the formation conductivity affects the initial value of the real part signal. When proppant conductivity and formation conductivity differ significantly, the effect of formation conductivity is negligible. Similar to the results in the previous simulations, the larger the transmitting-receiving distance, the smaller the amplitude of the real part signal. In addition, the imaginary part is less stable than the real part.

It can be illustrated from the figures in Section 3.1.3 that: (1) Changing the frequency of the subarray can effectively change the signal amplitude of the fracture location. If the signal amplitude of the coil system with a long transmitting-receiving distance is low, increasing the frequency of the coil can enhance the signal and make it clearer. However, in theory, higher frequencies would result in lower penetration, this means that the high frequencies cannot detect fractures over long distances. (2) The imaginary part signal with high frequency and long transmitter-receiving distance can also identify the location of the fracture.

Results can be obtained by analyzing Figure 8: (1) The voltage real part signal is very sensitive to the change of fracture conductivity. There is a significant amplitude increase at the location of the fracture, and the maximum value is reached at the center of the fracture. It is verified that the voltage real part signal can be used as the primary research data of this method. (2) Subarray 1 has the largest measured amplitude, and subarray 7 has the largest response range to fractures. (3) The imaginary part signal only produces a strong response to the material with high conductivity in subarray 7.

In Section 3.2.1, the presence of circular propped fracture does not influence the voltage imaginary part signal much, as Figure 10 shows. However, each subarray's voltage real part signal curves will show obvious signal mutation regions in the curves due to the secondary electromagnetic field interference caused by fracturing fractures. Therefore, the voltage real part signal curve is more sensitive to the presence of conductive proppants-filled fractures, and the voltage real part signal is chosen as the main research object of the electromagnetic signal response.

The results of the Section 3.2.2 are as follows: (1) The voltage real part signal can reflect the change of fracture size under various conditions, and the real part of the voltage is more sensitive than the imaginary part. (2) Different transmitting-receiving distance of subarrays has a different resolution to fracture size, and the maximum diameter of the circular model that can be resolved is different. A short transmitting-receiving distance is more able to distinguish thin-layer fractures, and a long transmitting-receiving distance is suitable for larger-size fractures.

Figure 13 indicates that the cross-sectional geometry of the model has little influence on the voltage real part signal of each sub-array when the total volume difference of fracture support is not large.

## 5. Conclusions

In this paper, we conducted a forward model based on the magnetic hydraulic fracture monitoring method and verified the feasibility of the electromagnetic method. We also provided an idea to interpret these signals. In particular, we discussed the results of forward modeling and reached the following conclusions:

1. In the signal curve measured by the tool proposed in this paper, the real part of the voltage signal is sensitive to various parameter changes, so it has more research value than the imaginary part signal.
2. The tool proposed in this paper can monitor the fracture location, and there are obvious signal fluctuations in all amplitude curves. Its specific signal expression form is as follows: The detected signal increases as it approaches the fracture and reaches its maximum when it reaches the center of the fracture. In addition, according to the different transmitting-receiving distances of the subarray, the more points of the maximum signal, the closer the shape of the ladder will be in the figure.
3. The proposed tool can theoretically ignore the influence of information factors, performs well in the longitudinal resolution of model thickness, and has a strong sensitivity to the volume of the whole model.

In the future, we plan to conduct parameter inversion research on the functional relationship between signal amplitude and fracture volume and find a new subarray configuration to reflect the cross-section shape of the fracture.

**Author Contributions:** Conceptualization, L.Z.; Methodology, L.Z.; Software, X.Z. and J.S.; Validation, X.Z. and C.C.; Formal analysis, X.Z. and P.J.; Investigation, P.J. and Z.L.; Data curation, J.S.; Writing—original draft, X.Z.; Writing—review and editing, L.Z.; Project administration, L.Z. All authors have read and agreed to the published version of the manuscript.

**Funding:** This research was funded by the National Natural Science Foundation of China under Grants 51874335, 52274057, and 52074340; the Major Scientific and Technological Projects of CNPC under Grant ZD2019-183-008; the Major Scientific and Technological Projects of CNOOC under Grant CCL2022RCPS0397RSN; the Science and Technology Support Plan for Youth Innovation of University in Shandong Province under Grant 2019KJH002, 111 Project under Grant B08028.

**Data Availability Statement:** Not applicable.

**Acknowledgments:** The authors would like to thank the Smart oil field Research group of China University of Petroleum (East China) for their assistance in numerical simulation.

**Conflicts of Interest:** The authors declare no conflict of interest.

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
