# Peer review of "Simulation Study for Hydraulic Fracture Monitoring Based on Electromagnetic Detection Technology"

_water, doi:10.3390/w15030568_

Round 1

Reviewer 1 Report

In this paper the authors practically try a forward modelling based on EM method for hydraulic fracture determination. Although there is extensive presentation of the experiment, I propose major revisions, mostly due to the language editing and references. My remarks are the following:

1. The MS needs extensive English editing in my opinion since the syntax is not good. Especially the abstract is very difficult to read and understand. There are more parts, in the rest of the manuscript, that are not easy to understand (e.g. lines 70-77, 385-389)

2. The introduction part and especially page 2 has not the appropriate style for references in text, as stated in the journal's instructions. The citations must be written as numbers based on the sequence of their citation and not with the usual way of mentioning the authors and publication date.

3. The final reference list has problems. Almost all journals’ references do not mention the journal’s name or they mention them as abbreviations joined with the last author’s name. Please study carefully the journal’s instructions regarding the way you should write the references.

4. Figure 1 is not mentioned in the text. It might also needs some improvements and explanations. 

6.  Line 90: Could you please provide references and justification for choosinh to use HDIL?

7. I believe that there should be a general discussion section, where you should move most of the ending parts of each sub-section in section 3. For example, lines 188-194, 225-230, 252-259, 288-294, 332-337, 365-371.

8.  Line 234:  what do you mean “…… the electromagnetic response would be got”?

9.  Can you please justify why subarrays 1,4,7 were selected?

10. Lines 390-392: Can you please add more details supporting your statement?

11. Please complete the "Author Contributions" and "Acknowledgments" (if applicable)

Reviewer 2 Report

The manuscript establishes the fracture model and the simplified electromagnetic detection tool model by FEM. In essence, the model is to solve the superposition of the electromagnetic field excited by the coil and the induced electromagnetic field of other media at different positions. Then it is intended to verify the effectiveness of this method according to the variation of superposition field potential.

While the idea behind this new framework is intriguing, but it needs very significant explanation and improvement before the paper is accepted for publication. Please see my comments below.

## Comments 1:This article is named “Method for Hydraulic fracture Diagnosis Based on Electro-magnetic Detection Technology”. But this paper mainly introduces a forward modeling method and data. The title of the article is informative but not necessarily relevant. It is suggested to modify the title to make it more attractive for the readers and for its possible citation.

## Comments 2:The authors briefly describe what is already known about the topic, but the research question is not outlined and justified given what is already known about the topic. Authors are suggested to improve the introduction of their article.

Authors must spend much more time writing their articles, especially when there are too many authors.

## Comments 3:In selection 2 “Mathematics Physics Model”, the process of the proppants or the filled fractures is not completely clear, many things are missing to adequately illustrate and define.

## Comments 4:In line 162, “Data were acquired every 0.1m as the tool moved.” That means the step size is 0.1m. In Table 2, the fracture thickness is set to 0.005m. The setting of step size may lead to missing critical signals. Please ensure the setting of step size is correctly.

## Comments 5:In line 304, “In order to simplify the calculation and present the results, the electromagnetic signal responses of representative subarrays 1, 4, and 7 are selected as the main results.” The author would do well to give reasons for his choice.

## Comments 6:In section 3.2.2, this chapter aims to show that shape has little effect on the signal. But we know that squares and circles are very similar. Please add more experiments with different shapes to support the author's point.

Round 2

Reviewer 1 Report

The authors have improved the manuscript.  There is only one minor correction to be done:

1.       The introduction part and especially page 2 has not been corrected and has not the appropriate style for references in text, as stated in the journal's instructions. The citations must be written as numbers based on the sequence of their citation. You should not mention the name of the authors of the references in the text.

For example, I provide you an example of the first sentence. The same should be applied to almost all the introduction part. 

You write: “Soumyadipta Sengupta proposed in 2012 that ferrofluid injection into fracture sup-42 port fracture makes fracture become a magnetic medium with electromagnetic properties 43 significantly different from the background reservoir[21]”.

It should be somehow like this:

“The author of [21] proposed in 2012 that ferrofluid injection into fracture sup-42 port fracture makes fracture become a magnetic medium with electromagnetic properties 43 significantly different from the background reservoir”

Please check all the introduction part regarding this matter.

Author Response

Thank you very much for your suggestions and comments. Those comments are all valuable and very helpful for revising and improving our paper. I’m going to write my response one by one as following:

Comment 1: The introduction part and especially page 2 has not been corrected and has not the appropriate style for references in text, as stated in the journal's instructions. The citations must be written as numbers based on the sequence of their citation. You should not mention the name of the authors of the references in the text.

For example, I provide you an example of the first sentence. The same should be applied to almost all the introduction part. 

You write: “Soumyadipta Sengupta proposed in 2012 that ferrofluid injection into fracture sup-42 port fracture makes fracture become a magnetic medium with electromagnetic properties 43 significantly different from the background reservoir[21]”.

It should be somehow like this:

“The author of [21] proposed in 2012 that ferrofluid injection into fracture sup-42 port fracture makes fracture become a magnetic medium with electromagnetic properties 43 significantly different from the background reservoir”

Please check all the introduction part regarding this matter.

RESPONSE: We have revised this part of the article you mentioned.